# Feasibility of 3D-Printed Locking Compression Plates with Polyether Ether Ketone (PEEK) in Tibial Comminuted Diaphyseal Fractures

**DOI:** 10.3390/polym15143057

**Published:** 2023-07-16

**Authors:** Hyung-Jin Chung, Ho-Beom Lee, Kwang-Min Park, Tae-Gon Jung, Sang-Bum Kim, Byoung-Gu Lee, Wan-Chin Kim, Jeong-Kil Lee

**Affiliations:** 1Department of Orthopaedic Surgery, Chungnam National University Sejong Hospital, Chungnam National University School of Medicine, 20, Bodeum 7-ro, Sejong-si 30099, Republic of Korea; leecomet@hanmail.net (H.-J.C.);; 2School of Mechanical Engineering, Chungnam National University, 99 Daehak-ro, Yuseong-gu, Daejeon 34134, Republic of Korea; 3Medical Device Development Center, Osong Medical Innovation Foundation 123, Osongsaengmyeong-ro, Osong-eup, Heungdeok-gu, Cheonju-si 28160, Chungbuk, Republic of Korea; 4Department of Mechanics-Material Convergence System Engineering, Hanbat National University, 125, Dongseo-daero, Yuseong-gu, Daejeon 34158, Republic of Korea

**Keywords:** 3D printing, polyether ether ketone locking compression plate (PEEK LCP), tibial fracture, bending test, axial compression test, axial torsion test

## Abstract

The applicability of a polyether ether ketone locking compression plate (PEEK LCP) fabricated using FDM (fused deposition modeling)-based 3D printing to treat actual patients was studied. Three different tests—bending, axial compression, and axial torsion—were conducted on tibial non-osteoporotic comminuted diaphyseal fracture samples fixed with the commercial titanium alloy LCP and 3D-printed PEEK LCP. Comparing the outcomes of these tests revealed that the commercial titanium alloy LCP underwent plastic deformation in the bending and axial torsion tests, though the LCP did not fail even when an external force greater than the maximum allowable load of the tibia fixture of the LCP was applied. Elastic deformation occurred in the 3D-printed PEEK LCP in the bending and axial torsion tests. However, deformation occurred even under a small external force, and its stiffness was 10% compared to commercial titanium alloy LCP. Thus, 3D-printed PEEK LCP can be applied to the fracture conditions in non-weight-bearing regions. The experimental results reveal detailed insights into the treatment of actual patients by considering the stiffness and high toughness of 3D-printed PEEK LCP.

## 1. Introduction

Metal implants are often used for stable fixation of bones after fractures or during orthopedic surgery. Metal implants with high stiffness are used to obtain absolute stability. However, implants with a relatively high stiffness compared to the bone limit interfragmentary motions may hinder secondary bone healing [1]. Secondary bone healing is induced via interfragmentary motion of a few millimeters, with the optimal range of micromotion being 0.2–1.0 mm [2,3,4]. Considering that metal implants have a larger elastic modulus than cortical bone, micromotions may result in increased callus formation. As such, several efforts have focused on identifying such plate materials in many orthopedic areas.

Polyether ether ketone (PEEK), which is a super engineering plastic, possesses excellent wear resistance and anti-corrosion properties. Further, as it is a bio-inactive material, it does not cause harmful reactions in human tissues or release harmful components. Furthermore, owing to characteristics such as natural radiolucency and MRI compatibility, PEEK usage as a biomaterial for orthopedic implants has gradually increased [5,6,7]. The Young’s modulus of PEEK is approximately 3–4 GPa, which is smaller than that of cortical bone, whereas that of carbon fiber reinforced PEEK is about 18 GPa, which is in the range of the cortical bone, i.e., 11 GPa to 21 GPa. Thus, various studies have focused on orthopedic surgical applications based on traditional manufacturing methods, such as direct processing of PEEK, compression molding, or injection molding [8,9]. However, implementing complex shapes using the direct processing method is difficult. While the PEEK processing method using molding caters to mass production in the future, it has facilitated the need for suitable molds.

3D printing technology yields a three-dimensional shape layer by layer with hundreds to thousands of layers of materials in the form of filament, powder, or resin. Thus, 3D printing, which is an additive manufacturing method, enables the formation of complex three-dimensional shapes that cannot be realized via manufacturing methods based on traditional machining [10,11,12]. These advantages have resulted in increased adoption of approaches that help plan surgery through a 3D-printed model that scans CT or MRI images before surgery, such as correcting complex fractures or deformities [13,14]. Considering their ability to create a patient-specific bone support mechanism by quickly scanning the patient’s bone structure and fracture state, 3D-printed PEEK implants have been widely used in various medical fields to improve patient satisfaction and safety and enable rapid recovery [14,15]. Moreover, 3D-printed PEEK scaffolds have proven beneficial in bone regeneration because they enable cell adhesion and proliferation by creating macropores in the implant’s structure [16,17].

The selective laser sintering (SLS) method supplies molding materials as powder and applies high-power laser energy to the powder particles to form shapes through sintering between powder particles. SLS can realize highly sophisticated and high-strength structures among 3D-printing technologies within a short dose time when applied to each layer. The compact nature of the laser spot induces sintering [18,19,20]. However, for PEEK, which possesses good mechanical properties and has drawn significant attention as a bio-compatible super engineering plastic, the powders must be heated to at least 250 °C to induce sintering, which requires a high-power laser [21]. As commercial SLS PEEK 3D printers are expensive and molding conditions must be precisely controlled for each output model in consideration of shrinkage and deformation that inevitably occur during work, the use of this approach in orthopedic surgical applications has not been extensively studied.

Fused deposition modeling (FDM) uses PEEK supplied through a spool in filament form, with PEEK 3D printing implemented via heating and melting at 350–400 °C near the extrusion part. The benefits of the PEEK FDM compared to the SLS process are its highly stable manufacturing conditions and negligible shrinkage that occurs afterwards. However, the disadvantage of FDM 3D printing is that since the laminating speed is quite slow, the material of the stacked layer is solidified until the next layer is formed. Therefore, the adhesive strength between layers is poorer than powder bed fusion 3D printing. Despite the drawbacks of the FDM method, the production of PEEK material medical devices using the FDM method has been extensively researched [22,23,24,25]. Honigmann et al. [22] verified the possibility of manufacturing various forms of patient-specific surgical implants, such as osteosynthesis plates and fragment osteosynthesis plates, through the FDM method and using PEEK materials. Limaye et al. [23] produced bar and tensile specimens of PEEK materials through fused deposition modeling (FDM) and evaluated their mechanical properties and bio-adhesion characteristics. In previous studies, the advantages of using PEEK for manufacturing implants for use in surgery were discussed [22,23].

This study aimed to produce a locking compression plate (LCP) with PEEK materials and using an FDM 3D printer. This study has significant differences from previous research into the application of PEEK LCP. The key distinction lies in the utilization of the FDM method to fabricate the PEEK LCP and evaluate its mechanical properties at a specific fracture site. In this study, a commercially available LCP model, which is commonly used for tibial non-osteoporotic diaphyseal fractures, was 3D printed using PEEK materials and an FDM 3D printer. The mechanical properties were evaluated under simulated conditions that resembled actual fracture site fixation. Additionally, the feasibility of applying the 3D-printed PEEK LCP was examined by comparing the evaluation results to those of the commercially available LCP. Furthermore, finite element (FE) analysis, which considered the experimental conditions, was conducted to analyze the differences in mechanical properties between the FDM-printed PEEK LCP and an ideal-shaped PEEK LCP.

## 2. Material and Methods

In the study, a workflow, including material preparation, experimental procedures, and analysis was followed, as shown in Figure 1. Firstly, a commercially available LCP model commonly used in tibial fracture surgery was selected, and a three-dimensional model of the commercial LCP was obtained through reverse engineering. Subsequently, an FDM machine capable of implementing melt extrusion modeling with high-melting-point plastic material and PEEK filament was chosen, and the reverse-engineered LCP shape was fabricated using these materials. Next, the fabricated 3D-printed LCP was applied to the fixation part of the actual tibial fracture site to evaluate its mechanical properties, and FE analysis was performed by considering the same experimental conditions.

### 2.1. Material Preparation

A titanium alloy LCP applicable to a tibial non-osteoporotic comminuted diaphyseal fracture was selected to fabricate the PEEK LCP. The titanium alloy plate was an off-the-shelf product (8-hole, 4.5-mm narrow LCP), which we procured from a company that specializes in the production of orthopedic implants (Synthes GmbH, Oberdorf, Switzerland). The length of LCP was 152 mm, and it had eight holes. The eight holes were symmetrically placed in groups of four around the center of the LCP. A 3D scanner (ATOS Compact Scan, GOM, Braunschweig, Germany) was used to obtain a 3D model, and the LCP shape was reverse engineered. Figure 2 shows the selected titanium alloy LCP and the reverse-engineered model.

To print the reverse-engineered LCP with PEEK material, 3D printing was attempted using PEEK filament that contained 10% carbon fiber, which is known to possess the strength of cortical bone. However, as reported in previous studies, the material was highly brittle. Moreover, considering that the LCP model had a thickness of 5 mm, it was difficult to test via attachment to a sawbone because it broke even under a small bending force [26]. Thus, among general PEEK materials, a filament with excellent output characteristics was selected. Table 1 shows the material properties of the PEEK filament (IEMAI PEEK 3D Filament, IEMAI, Dongguan, China) used for 3D printing.

The filament, which had a diameter of 1.75 mm, was printed using a PEEK-only 3D printer (MAGIC-HT-PRO, IEMAI, Dongguan, China). Table 2 shows the output conditions for LCP. Internal filling was performed at 100%, with the nozzle temperature set to approximately 70 °C higher than the melting point of PEEK, i.e., 343 °C, for sufficient extrusion speed and layer settling. The output bed was set near to the glass transition temperature of PEEK, i.e., 143 °C, to minimize thermal shrinkage when the bed was seated, while the temperature inside the chamber was set to 90 °C to support the thermal shrinkage and molded layer. Figure 3 shows the printed PEEK LCP. The size, thickness, and width of the printed shape were 152.4 ± 0.2 mm, 4.92 ± 0.05 mm, and 13.6 ± 0.1 mm, respectively, with almost no shape error compared to the original shape of the titanium alloy LCP.

### 2.2. Specimen Preparation

The titanium alloy plate and the 3D-printed PEEK plate were fixed under comminuted fracture conditions to examine the mechanical properties of the commercial titanium alloy LCP and the 3D-printed PEEK LCP under tibial non-osteoporotic comminuted diaphyseal fracture conditions, which was similar to the conditions of previous studies. Next, the results of bending, axial compression, and axial torsion tests were obtained [27,28]. Locking screws, which were provided by a commercial LCP manufacturer, were used to fix the LCP to the sawbone tibia. A titanium alloy plate and a PEEK plate were fixed to a pre-manufactured sawbones tibia (#3401, Sawbones, Pacific Research Laboratories, Inc., Vashon, WA, USA) using six locking screws. Similar to the previous experiment, a 10-millimeter fracture gap was created in the tibial diaphysis under comminuted fracture conditions [28,29]. Two holes in the middle, which were close to the fracture gap, were emptied, and locking screws were fastened to the remaining six holes using a torque driver with a force of 4 N-m. The plate was elevated by 1 mm in the tibia model, with biological fixation performed to preserve periosteal perfusion. Figure 4a shows the titanium alloy LCP and 3D-printed PEEK LCP fixed to the sawbone tibia. As shown in Figure 4b, the bending test was conducted without fixing both ends of the sawbone. To ensure uniform load during the axial compression and axial torsion tests, both ends of the sawbone tibia were fixed using an experimental jig and self-polymerizing resin (Vertex Trayplast NF, Vertex Dental BV, Zeist, The Netherlands), as shown in Figure 4c.

### 2.3. Methods for Mechanical Tests

We aimed to compare and analyze the biomechanical properties of tibial non-osteoporotic comminuted diaphyseal fracture samples fixed with commercial titanium alloy LCP and 3D-printed PEEK LCP. Therefore, bending, axial compression, and axial torsion tests were performed using a universal testing machine (MTS E45, MTS Systems, MN 55344-2247, Eden Prairie, MN, USA). For each test, two samples were prepared for each LCP type. Therefore, a total of 12 sawbone tibia were used.

For the bending test, four-point bending was applied to generate a constant bending moment over the plate’s entire length. The experiment was performed according to the ASTM F382 standard, with the center span and loading span distances set to 70 mm and 55 mm, respectively. The test specimen was placed flat on the lower specimen, with the fractured part placed at the center, and a compressive load was applied at a rate of 5 mm/min until the specimen fractured or the tibias at both ends came in contact with each other. Load–displacement data were recorded at 50 Hz during the experiment.

For the axial compression test, the specimen was held vertically, with a compressive load applied at a rate of 5 mm/min until the specimen was fractured or the tibias at both ends came in contact with each other. The load–displacement data were recorded at 50 Hz. Moreover, for a torsion test, torsional torque was applied at the top of the fixed specimen at a rate of 0.1°/s while the specimen was held vertically for the axial torsion test.

## 3. Data and Results

### 3.1. Experimental Results

Figure 5 shows the state of the specimen after the bending test and the resulting displacement–load curve. As shown in Figure 5, no fracture was observed in the LCP until the point where the distal and proximal tibias were in contact with each other in both types of specimens. Factors such as stiffness, yield load, bending structural stiffness, and bending strength were calculated using the load–displacement curve presented in Figure 5, with the results summarized in Table 3. The bending structural stiffness and bending strength were calculated according to the ASTM F382 standard, considering both the center span distance and loading span distance.

As shown in Table 3, the stiffness of the 3D-printed PEEK LCP against the bending load was 8.7%, and the yield load and bending strength were 10.4% compared to the commercial titanium alloy LCP. It is noteworthy that, unlike commercial titanium alloy LCPs, 3D-printed PEEK LCPs did not undergo plastic deformation.

Figure 6 shows the image taken after the axial compression test for each case and the load–displacement result. In the case of the 3D-printed LCP, buckling of the LCP occurred; thus, the load was applied until the tibias at both ends were in contact. However, the LCP was not broken at the end of the test. In the case of commercial titanium alloy LCP, its stiffness on the axial compression was greater than the stiffness of other supporting parts, including the sawbone tibia. Thus, the test was conducted until the sawbone tibia was fractured. Table 4 shows stiffness, maximum displacement, and maximum load.

The results in Figure 6 and Table 4 show that, on average, the compression load of the 3D-printed PEEK LCP continuously increases until the compression displacement reaches 1.65 mm, and the maximum allowable load is about 770 N. When commercial titanium alloy LCP was applied, almost no deformation of LCP and tibia fracture occurred. Therefore, the stiffness, the maximum displacement, and the maximum load listed in Table 4 can be regarded as mechanical properties for axial compression of the tibia structure that is fixed with the commercial titanium alloy LCP. According to the values in Table 4, the 3D-printed PEEK LCP exhibited a maximum axial compression load of 20% and a stiffness of 45% of that of the tibia structure fixed with the commercial titanium alloy LCP.

During the torsion test, in the 3D-printed PEEK LCP, no fracture occurred, with only deformation occurring; thus, torque was applied until a torsion angle of approximately 100° was achieved. The commercial titanium alloy LCP separated from the tibia at a torsion angle of approximately 26°. Table 5 shows the stiffness, maximum angular displacement, and maximum torque calculated based on the torque-angle data in Figure 7.

The results in Table 5 indicate that the maximum allowable torque and stiffness of the 3D-printed PEEK LCP were approximately 29% and 11%, respectively, compared to the tibia fixture fixed using the commercial titanium alloy LCP.

### 3.2. Finite Element Analysis

As mentioned in the Introduction section, when manufacturing a 3D mechanical object using an FDM method, the adhesion between layers may be lowered due to the temporal difference between consequential layers. In this section, the ideal shape is modeled using IEMAI’s filament material, which is a material used for 3D-printed PEEK LCP, and its bending, compression, and torsion characteristics are analyzed for comparison with the experimental results. For finite element analysis, general-purpose mechanical analysis software ANSYS 2022 R2 (ANSYS, Inc., Canonsburg, PA, USA) was used, and the mechanical properties shown in Table 6 were considered for PEEK material and cortical bone. In the analysis, the simplified cortical bone model was applied.

The LCP shape was set as a “bond” constraint condition by vertically fastening three screws to the cortical bone tibia for analysis under conditions similar to the experimental conditions, as shown in Figure 4a. For the bending analysis, the analysis model was defined to have the support and load applications shown in Figure 4b. This analytical setup is shown in Figure 8a. The safety factor and deformation of the LCP shape were examined while increasing the load from 0 N to 120 N in fixed increments of 10 N added to the load-applied part. A model with both ends fixed to the cortical bone tibia was constructed for the compression and torsion analyses, as shown in Figure 8c. For the compression analysis, the safety factor and deformation of the LCP shape were reviewed by increasing the load on the upper part from 0 N to 1000 N in fixed increments of 100 N. For torsion analysis, the safety factor and deformation of the LCP shape were examined while increasing the torque applied at the upper part in increments of 0, 1, and 10 N-m.

Figure 8b,d,e shows the deformation shape of the model when the maximum load or maximum torque was applied during the bending, compression, and torsion analyses. Based on the bending test results of the 3D-printed PEEK LCP shown in Figure 5, the tibias at both ends came into contact at the point where the average yield load was 77 N and the bending displacement was 12 mm. Based on the bending analysis of the LCP with an ideal shape and made the same material, a safety factor smaller than 1 at 80 N showed yield characteristics. When the maximum load of 120 N was applied, the bending displacement was 14.53 mm. The compression test results of the 3D-printed PEEK LCP shown in Figure 6 confirmed that the average yield load was 770 N and the compression displacement was approximately 1.65 mm. The compression analysis result of the LCP with the same material and ideal shape indicated that a safety factor less than 1 at 900 N exhibited yield characteristics. The torsion test results of the 3D-printed PEEK LCP in Figure 7 showed that it possessed a non-linear angular displacement with respect to the applied torque. The maximum applied torque was 7.2 N-m as an average value. However, as the torque range shows a linear relationship between the applied torque and the angular displacement within 1.6 N-m, the yield torque can be regarded as 1.6 N-m. Based on the torsion analysis of the LCP with the same material and ideal shape, the applied torque showed a linear torsion angle up to 1.7 N-m, while the deformation angle was 145° when the maximum torque of 10 N-m was applied. Table 7 shows the direct comparison between experimental results and FEA analysis results for each test.

## 4. Discussion

The 3D-printed PEEK LCP fabricated in this study is not recommended for tibial non-osteoporotic diaphyseal comminuted fracture conditions. The stiffness is significantly lower than that of the commercial titanium alloy LCP, and the maximum permissible axial compression load is 770 N. Thus, it cannot withstand the load applied by an adult of average weight. However, considering the stiffness level and high toughness characteristics of the 3D-printed PEEK LCP confirmed in this experiment, it has the potential for use in upper extremity fractures.

Schliemann et al. [30] reported that the CFR-PEEK plate possessed a lower fixation strength and greater motions than the titanium alloy plate for unstable proximal humerus fractures in laboratory experiments. However, for proximal humerus fractures in real patients, better results were reported in terms of clinical progress and loss of reduction when using CFR-PEEK plates than using metal plates [31]. Tarallo et al. [32] reported that the CFR-PEEK plate in a distal radius fracture showed a good post-operative clinical outcome after 12 months. The complications observed after more than four years of follow-up were similar to those of other metal plates, though intraoperative plate rupture occurred in 4% of cases [33]. In all cases, commercial CFR-PEEK plates sourced from a specialist company were used to produce orthopedic implants. The PEEK plate applied to the proximal humerus and distal radius possessed a smaller length and a larger width than the 3D-printed PEEK plate fabricated in this study. Thus, it was expected to be relatively strong against bending forces compared to the 3D-printed PEEK plate fabricated in this study. In the upper extremities, axial compression would have been smaller after surgery, and the use of an orthosis, such as a splint, would not have significantly impacted the axial torsion. Thus, the clinical results of the PEEK plate applied to the upper extremity could have been desirable.

The finite element analysis results indicate that the bending and compression yield strengths of the 3D-printed PEEK LCP were 96% and 86%, respectively, compared to PEEK LCP with an ideal shape, and the yield torque against torsion was 94%. Compared to the mechanical properties of the ideal model, the mechanical properties of the LCP manufactured via the FDM method are judged to be at a similar level in the cases of bending and torsion. However, in the case of the compression test, degradation over 10% on yield strength occurs. This result occurs because when PEEK material is processed via the FDM method, the laminated layer cools and hardens during the time required for the next layer to be laminated after one layer is laminated. The direction of the force applied to the 3D-printed PEEK LCP during bending and torsion tests is the same as the direction of layering during FDM 3D printing. However, during compression testing, the direction of the applied force is orthogonal to the direction of layering, resulting in a further reduction in yield strength.

PEEK is used as a bone fusion material in many orthopedic surgeries. With the popularization of 3D printing in this industry, it has become possible to produce patient-specific 3D-printed PEEK plates. This development allows the production of patient-specific surgical guides and orthopedic implants through collaborative design with manufacturing companies. However, to date, these processes have mainly been implemented using specialized equipment that requires high-powered lasers, such as the multi-jet binding technique or SLS technology, along with skilled operators and specific workflows. As a result, many hospitals face difficulties in establishing all-in-one hospital systems due to the need for expensive and specialized 3D-printing equipment and skilled personnel. This study reports the ability to obtain mechanical properties at the material level through the production of PEEK plates via a cost-effective and widely used FDM 3D-printing method. PEEK material offers higher mechanical characteristics than other FDM materials, allowing its application in various customized surgical guides. Additionally, as proposed in this study, it can be utilized as an orthopedic implant for surgeries on non-weight regions. Therefore, when producing patient-specific surgical guides and orthopedic implants in hospitals, it is expected to alleviate the economic and technical burden of acquiring specialized and expensive 3D-printing equipment and skilled personnel.

## 5. Conclusions

A feasibility study was conducted on tibial non-osteoporotic comminuted diaphyseal fracture specimens fixed via a PEEK LCP fabricated using FDM 3D printing. We conducted bending, compression, and torsion tests on a FDM 3D-printed PEEK LCP fixed to a sawbone tibia. The yield bending load, maximum compression load, and maximum allowable torque were 77 N, 770 N, and 7.2 Nm, respectively. These values were 10.4%, 20%, and 29% of the experimental results obtained for a commercially available titanium LCP model fixed to the sawbone tibia with the same shape. Considering the weight of an average adult and the load applied to the leg when walking, it is not suitable for use in the tibial non-osteoporotic comminuted diaphyseal fracture condition. However, considering the stiffness level and high toughness characteristics of 3D-printed PEEK LCP confirmed through this experiment, it may be suitable for fracture conditions in non-weight-bearing regions, such as clavicle, humerus, ulna, radius, etc.

## Figures and Tables

**Figure 1 polymers-15-03057-f001:**
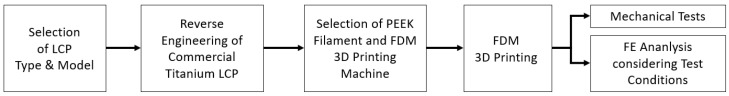
Workflow of the study, extending from material preparation to methods, including experiments and analysis.

**Figure 2 polymers-15-03057-f002:**
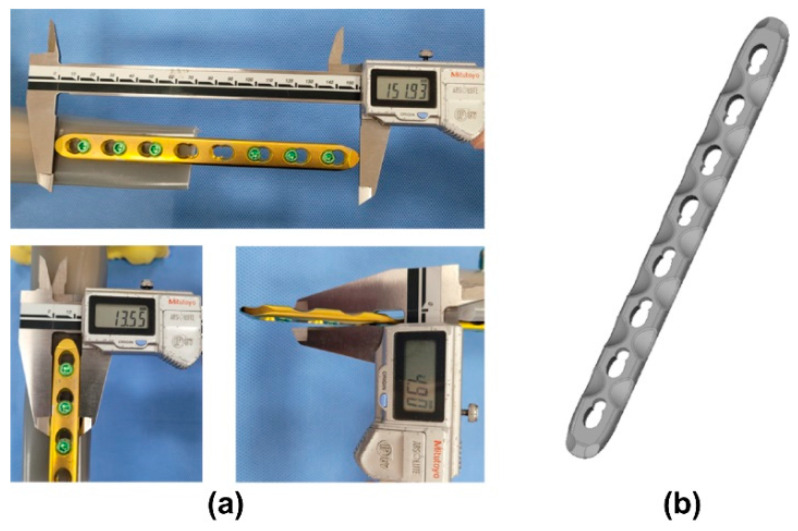
Selected titanium alloy LCP with eight holes (**a**) and its scanned 3D model (**b**).

**Figure 3 polymers-15-03057-f003:**
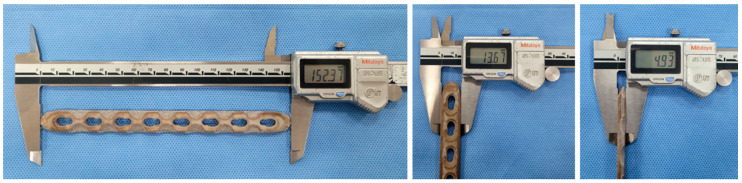
A 3D-printed PEEK LCP.

**Figure 4 polymers-15-03057-f004:**
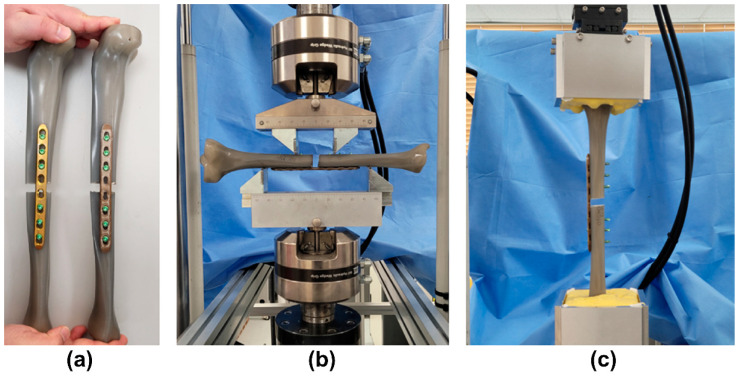
Specimens prepared for evaluation of mechanical properties and specimen preparation in each test: (**a**) titanium alloy LCP fixed tibia (left) and 3D-printed PEEK LCP fixed tibia (right); (**b**) specimen prepared for bending test; (**c**) specimen prepared for compression and torsion tests.

**Figure 5 polymers-15-03057-f005:**
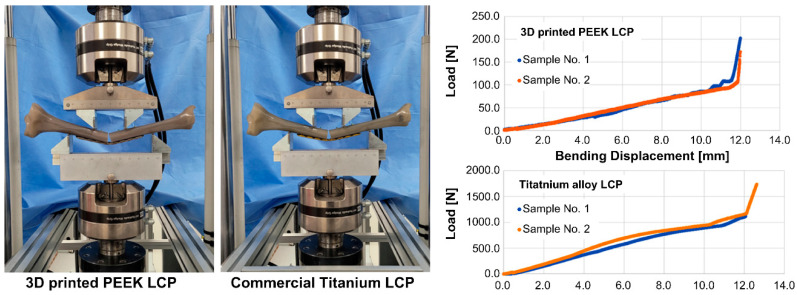
Bending test pictures and displacement–load curve measurement results of structures using 3D-printed PEEK LCP and commercial titanium alloy LCP.

**Figure 6 polymers-15-03057-f006:**
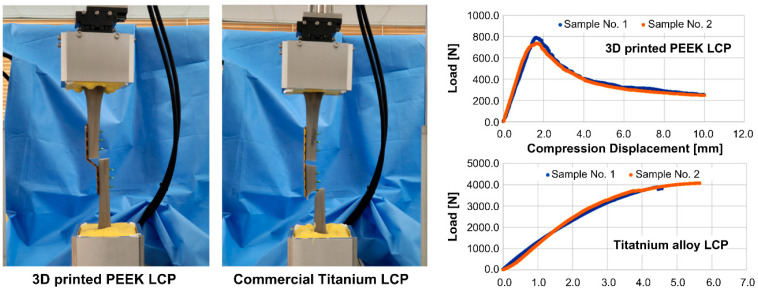
Axial compression test setup and load–displacement curve measurement results of structures to which the 3D-printed PEEK LCP and commercial titanium alloy LCP were applied.

**Figure 7 polymers-15-03057-f007:**
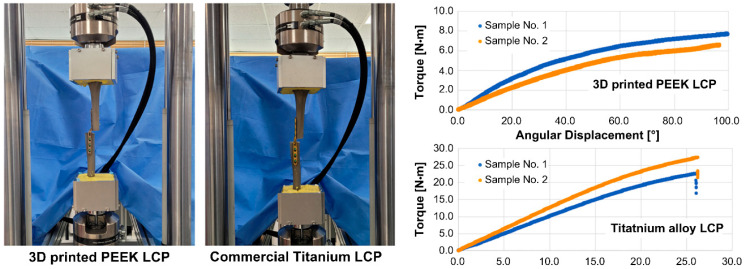
Axial torsion test setup and load–displacement curve measurement results of structures using 3D-printed PEEK LCP and commercial titanium alloy LCP.

**Figure 8 polymers-15-03057-f008:**
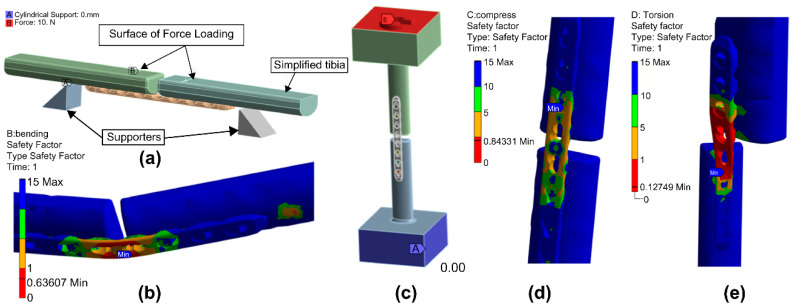
Results of bending, compression, and torsion analysis models under maximum load or maximum torque. (**a**) Bending analysis model, (**b**) bending analysis result safety factor of 0.6361, (**c**) compression and torsion analysis model, (**d**) compression analysis result safety factor of 0.8433, and (**e**) torsion analysis result safety factor of 0.1275.

**Table 1 polymers-15-03057-t001:** Mechanical and thermal properties of PEEK filament applied in 3D printing.

Item	Condition	Methods	Value
Tensile strength	Yield, 23 °C	ISO 527	100 MPa
Flexural strength	Yield, 23 °C	ISO 178	170 MPa
Flexural modulus	23 °C	ISO 178	4.2 GPa
Compression strength	23 °C	ISO 604	125 MPa
Melting point		ISO 11357	343 °C
Glass transition		ISO 11357	143 °C

**Table 2 polymers-15-03057-t002:** Printing condition for the fabrication of PEEK LCP.

Item	Condition
Print speed	35 mm/sec
Nozzle temperature	410 °C
Chamber temperature	90 °C
Bed temperature	130 °C
Infill	100%
Printing bottom direction	X-Y

**Table 3 polymers-15-03057-t003:** Calculated mechanical characteristics when applying 3D-printed PEEK LCP and commercial titanium alloy LCP in bending tests.

Characteristics	3D-Printed PEEK LCP			Commercial Titanium Alloy LCP		
	#1	#2	Average	#1	#2	Average
Stiffness (N/mm)	11.01	9.15	10.08	104.78	127.61	116.20
Yield load (N)	74.69	78.87	76.78	761.85	718.68	740.27
Bending structural stiffness (MN/mm^2^)	0.89	0.74	0.82	8.45	10.29	9.37
Bending strength (N/mm)	2054	2169	2111	20,951	19,764	20,357

**Table 4 polymers-15-03057-t004:** Calculated mechanical characteristics in both cases that applied 3D-printed PEEK LCP and the commercial titanium alloy LCP for an axial compression test.

Characteristics	3D-Printed PEEK LCP			Commercial Titanium Alloy LCP		
	#1	#2	Average	#1	#2	Average
Stiffness (N/mm)	533	586	560	1191	1314	1252
Maximum displacement (mm)	1.63	1.68	1.65	4.45	5.63	5.04
Maximum load (kN)	0.79	0.74	0.77	3.88	4.07	3.98

**Table 5 polymers-15-03057-t005:** Calculated mechanical characteristics in both cases of applying 3D-printed PEEK LCP and commercial titanium alloy LCP for axial torsion test.

Characteristics	3D-Printed PEEK LCP			Commercial Titanium Alloy LCP		
	#1	#2	Average	#1	#2	Average
Stiffness (N-m/deg)	0.15	0.11	0.13	1.02	1.25	1.13
Maximum angle (degree)	99.7	96.7	98.2	26.0	26.2	26.1
Maximum torque (N-m)	7.8	6.7	7.2	22.7	27.4	25.1

**Table 6 polymers-15-03057-t006:** Mechanical properties of materials used for simulations.

Material	Density (g/cm^3^)	Young’s Modulus (GPa)	Poisson’s Ratio	Tensile Strength (MPa)	Compression Strength (MPa)
IEMAI PEEK	1.3	3.76	0.39	100	125
Cortical bone	1.8	15	0.62	114	205

**Table 7 polymers-15-03057-t007:** Comparison between experimental results and FEA analysis results for each test.

	Bending	Compression	Torsion	
	Yield Load [N]	Yield Load [N]	Yield Torque [N-m]	Maximum Torque Applied [N-m]
FEA (ANSYS)	80	900	1.7	10
Experiment	77	770	1.6	7.2

## Data Availability

Data available on request due to restrictions, e.g., privacy or ethical.

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
