# Peer review of "Feasibility of 3D-Printed Locking Compression Plates with Polyether Ether Ketone (PEEK) in Tibial Comminuted Diaphyseal Fractures"

_polymers, 2023, doi:10.3390/polym15143057_

Round 1

Reviewer 1 Report

Introduction:

-     Please avoid sentences without references.

 -       "Honigmann et al. [20] verified the possibility of manufac-turing various forms of patient-specific surgical implants, such as osteosynthesis plates and fragment osteosynthesis plates, through the FDM method using PEEK materials. Li-maye et al. [21] produced bar specimens and tensile specimens of PEEK materials through fused deposition modeling "

-         I suggest summarizing previous studies results and\or conclusion, even if limited, outlining the current understanding in the literature, and set the context and point out the gap in knowledge that the rest of the paper will fill.

 Discussion:

 - “However, the results are considered reliable as similar experimental results were obtained for each specimen without significant differences.” Add references.

 -Please elaborate on the costs, cost-effectiveness, and economic considerations. Additionally, discuss whether the printing of PEEK can become an all-in-hospital process and provide insights into future considerations.

Author Response

Introduction:

  1.  Please avoid sentences without references.
  • Thank you for your considerable recommendation. We corrected the sentence according to reviewer’s comment.
  1. "Honigmann et al. [20] verified the possibility of manufac-turing various forms of patient-specific surgical implants, such as osteosynthesis plates and fragment osteosynthesis plates, through the FDM method using PEEK materials. Li-maye et al. [21] produced bar specimens and tensile specimens of PEEK materials through fused deposition modeling "

-         I suggest summarizing previous studies results and\or conclusion, even if limited, outlining the current understanding in the literature, and set the context and point out the gap in knowledge that the rest of the paper will fill.

  • Thank you for your considerable recommendation. We added the sentence according to reviewer’s comment.

 Discussion:

  1. “However, the results are considered reliable as similar experimental results were obtained for each specimen without significant differences.” Add references.
  • Thank you for your considerable recommendation. We confirmed and judged that the contents of the paragraph including the commented sentence are unnecessary to understand the results of this study. Therefore, we deleted the paragraph.
  1. Please elaborate on the costs, cost-effectiveness, and economic considerations. Additionally, discuss whether the printing of PEEK can become an all-in-hospital process and provide insights into future considerations.
  • Thanks to reviewer’s comment. According to reviewer’s suggestion we added the paragraph.

Reviewer 2 Report

We received paper from polymers with the title “Feasibility of 3D-printed locking compression plate with Polyether ether ketone (PEEK) in tibial diaphyseal comminuted fracture”. The paper was interesting on how to used PEEK as the compression plate for treatment of broken or crack in tibia bone. Several items need to be revised before it can be goes to further analysis.

1.       This claim “Thus, while 3D-printed PEEK LCP is not suitable for tibial non-osteoporotic diaphyseal comminuted fracture conditions, it may be used for fracture conditions in non-weight-bearing regions.” If this study is not suitable, then, it is valid or it is have novelty for the present study?

2.       The present study evaluated the used of PEEK. However, the last abstract claimed “However, additional research on the manufacturing method of 3D-printed plates or new materials is required.”. it that true that the present paper have not finished to evaluate PEEK in the 3D-printed locking compression plate?

3.       Keywords mechanical study is not influenced. Thus this term can be neglected.

4.       The term and the general applications of 3D printing as shown in the following section need to be added with more references “3D printing technology yields a three-dimensional shape layer by layer with hundreds to thousands of layers of materials in the form of filament, powder, or resin. Thus, 3D printing, an additive manufacturing method, enables the formation of complex three-dimensional shapes that cannot be realized in traditional manufacturing methods based on traditional machining [10,11].” The application of additive manufacturing are widely used in different applications. Thus, the following papers should be added in to the paper. In art, Excellent performance of hybrid model manufactured via additive manufacturing process reinforced with GFRP for sport climbing equipment.

5.       The claim of the present sub section such as “The selective laser sintering (SLS) method supplies moulding materials as powder and applies high-power laser energy to the powder particles to form shapes through sintering between powder particles. SLS can realize highly sophisticated and high-strength structures among 3D printing technologies within a short dose time applied to each layer. The compact nature of the laser spot induces sintering [17,18].” This should be added with the other application such as in high structures. The following references can be used to enrich the introductions, the AM can be used to produce PAF, First-rate manufacturing process of primary air fan (PAF) coal power plant in Indonesia using laser powder bed fusion (LPBF) technology.

6.       In the last paragraph of introduction, the gap between the previous study summarized in the introduction should be mentioned. Then, the novelty also need to be clearly stated.

7.       The present claimed “Internal filling was performed at 100%, with the nozzle temperature set to approximately 70 °C higher than the melting point of the material for sufficient extrusion speed and layer settling.” Is that any difference if the printing process used normal temperature and 70°C higher? And why the authors used 70 °C higher?

8.       Fig. 1 and Fig. 2 have no impact to the paper. The data related to the dimension of the metal CP that gained from the original manufactured and the results from AM, should be compared using bar graph or using table. Rather than measurement using calliper than take picture on it.

9.       We recommend to the authors to add 1 figure related to the schematic study from sample preparation to the characterization and evaluation in one figure.

10.   Why the authors want to used PEEK compared with Titanium as shown in Fig. 5 rather than using PLA or ABS or Nylon?

11.   What the version of ANSYS software that being used to validate the present study?

12.   In the conclusion, please add all the important results in one paragraph, in compact mode and straightforward, and stated in value mode. i.e. XX MPa.

Please carefully check all the entire manuscript and revised if the grammar have issues on it.

Author Response

Reviewer 2

  1. This claim “Thus, while 3D-printed PEEK LCP is not suitable for tibial non-osteoporotic diaphyseal comminuted fracture conditions, it may be used for fracture conditions in non-weight-bearing regions.” If this study is not suitable, then, it is valid or it is have novelty for the present study?
  • Thank you for your comment. One of our study’s strengths is that we made the PEEK LCP using 3D-print and compared the commercial LCP plate and 3D-printed PEEK LCP under various conditions in the specific tibial non-osteoporotic diaphyseal fracture. Many studies have been reported on surgical implants made of 3D printed PEEK. However, There was no study comparing commercial plate and PEEK plate under specific condition such as tibial non-osteoporotic fracture.
  • Based on our research findings, the significant strength of our study is the prospect that the 3D-printed PEEK LCP might potentially replace the commercial LCP plate currently used in clinical practice for fracture conditions in non-weight-bearing regions such as upper extremities. We clearly stated about this results in the “Conclusion”.
  • Furthermore, because of less previous studies, the combination of a biomechanical study using saw bone and finite element analysis in our research is regarded valuable.
  1. The present study evaluated the used of PEEK. However, the last abstract claimed “However, additional research on the manufacturing method of 3D-printed plates or new materials is required.”. it that true that the present paper have not finished to evaluate PEEK in the 3D-printed locking compression plate?
  • Thank you for your comment. PEEK material, which has mechanical strength 2 to 3 times that of nylon or ABS and has proven superior heat resistance and bio-compatibility, is currently being actively reviewed for its applicability to orthopedic surgery. In this study, the applicability of PEEK material processing using FDM 3D printing with high processing freedom was investigated for tibial fracture junction surgery. In order to consider this possibility, the same model as the titanium LCP model that is actually being applied was made of PEEK and various mechanical tests were conducted. In conclusion, it was confirmed that the mechanical properties that are difficult to apply to tibial fracture junction surgery when PEEK material is made of FDM. However, based on the results of experiments and analysis, it was concluded that it could be used for fracture bonding applications with low applied loads. The last sentence of Abstract was intended to express the need for overall research and experimental research on materials that have higher mechanical properties and can be 3D printed in the future. However, as the reviewer pointed out, we think there is enough potential for misunderstanding that this study is an unfinished study. So, delete that sentence.
  1. Keywords mechanical study is not influenced. Thus this term can be neglected.
  • Thank you for your considerable recommendation. We deleted the keyword.

  1. The term and the general applications of 3D printing as shown in the following section need to be added with more references “3D printing technology yields a three-dimensional shape layer by layer with hundreds to thousands of layers of materials in the form of filament, powder, or resin. Thus, 3D printing, an additive manufacturing method, enables the formation of complex three-dimensional shapes that cannot be realized in traditional manufacturing methods based on traditional machining [10,11].” The application of additive manufacturing are widely used in different applications. Thus, the following papers should be added in to the paper. In art, Excellent performance of hybrid model manufactured via additive manufacturing process reinforced with GFRP for sport climbing equipment.
  • Thank you for your considerable recommendation. Thank you very much for the opportunity to read the current article for us. We added the citation for reference.
  1. The claim of the present sub section such as “The selective laser sintering (SLS) method supplies moulding materials as powder and applies high-power laser energy to the powder particles to form shapes through sintering between powder particles. SLS can realize highly sophisticated and high-strength structures among 3D printing technologies within a short dose time applied to each layer. The compact nature of the laser spot induces sintering [17,18].” This should be added with the other application such as in high structures. The following references can be used to enrich the introductions, the AM can be used to produce PAF, First-rate manufacturing process of primary air fan (PAF) coal power plant in Indonesia using laser powder bed fusion (LPBF) technology.
  • Thank you for your considerable recommendation. Thank you very much for the opportunity to read the current article for us. We added the citation for reference.
  1. In the last paragraph of introduction, the gap between the previous study summarized in the introduction should be mentioned. Then, the novelty also need to be clearly stated.
  • Thanks to reviewer’s suggestion. We corrected it according to reviewer’s suggestion.
  1. The present claimed “Internal filling was performed at 100%, with the nozzle temperature set to approximately 70 °C higher than the melting point of the material for sufficient extrusion speed and layer settling.” Is that any difference if the printing process used normal temperature and 70°C higher? And why the authors used 70 °C higher?
  • Thank you for your considerable recommendation. The melting point of the PEEK material we applied is 343 °C, as shown in Table 1 and the text. In general, when FDM 3D printing PEEK material, the temperature of the extruder nozzle is set between 380°C and 430°C. However, there is a recommended extruder nozzle temperature according to the PEEK material manufacturer. IEMAI's PEEK filament applied in this study recommends 410 ~ 415°C for the melting point of PEEK material, which is 70°C higher than 343°C. Therefore, these conditions were applied when printing PEEK materials.

  1. Fig. 1 and Fig. 2 have no impact to the paper. The data related to the dimension of the metal CP that gained from the original manufactured and the results from AM, should be compared using bar graph or using table. Rather than measurement using calliper than take picture on it.
  • Thanks to reviewer’s comment. In many preceding studies examining the production of LCP as a material to replace titanium, there are often cases where a generally accurate LCP reference model is not presented. Or, in many cases, the shape of the model number presented is different from the actual shape. Therefore, we tried to accurately report which commercially available titanium LCP was applied in this study by accurately showing the photo and dimensions of the titanium LCP model as shown in Figure 1(revision version Fig 2) . In addition, the shape of the LCP printed with PEEK material in Figure 2 (revision version Fig 3) was accurately presented along with a photograph to emphasize that it was manufactured in the same size and shape as the titanium LCP through reverse engineering.
  1. We recommend to the authors to add 1 figure related to the schematic study from sample preparation to the characterization and evaluation in one figure.
  • Thank you for your comment. According to reviewer’s comment, we added a figure to express flow of the research.
  1. Why the authors want to used PEEK compared with Titanium as shown in Fig. 5 rather than using PLA or ABS or Nylon?
  • Thank you for your comment.We wanted to compare the characteristics of the actual LCP plate, which is made of titanium and commonly used in clinical practice, with the 3D-printed PEEK LCP
  • PEEK (Polyether Ether Ketone) is a commonly used thermoplastic polymer in the medical field due to its excellent mechanical properties, biocompatibility, and radiolucency. PEEK has been widely investigated and used in orthopedic implants and medical devices.
  • On the other hand, PLA (Polylactic Acid), ABS (Acrylonitrile Butadiene Styrene), and Nylon are also popular materials in 3D printing. However, they have different material properties compared to PEEK and may not possess the same level of mechanical strength, biocompatibility, or radiolucency required for certain medical applications.
  • Super engineering plastics such as PEEK or ULTEM, which have a melting temperature of over 300°C, have mechanical strength three to four times greater than conventional PLA/ABS, and excellent heat resistance. Unlike ULTEM, PEEK material has high biocompatibility, so many studies are being conducted to apply PEEK material with high strength to orthopedic procedures.

  1. What the version of ANSYS software that being used to validate the present study?
  • Thanks to reviewer’s comment. We used ANSYS 2022R2. We added it in the manuscript.
  1. In the conclusion, please add all the important results in one paragraph, in compact mode and straightforward, and stated in value mode. i.e. XX MPa.
  • Thanks to reviewer’s comment. We modified the conclusion according to reviewer’s comment.

Reviewer 3 Report

Comments to Authors:

As is stated in the Discussion section, the main lamination of the study was the small sample size of only 2 test specimens. It would be best to bolster that with more analysis of the FEA data. There is no table present discussing the results of the FEA data, only the initial parameters.

This manuscript needs to be carefully edited for grammar. Many corrections can be found below, but not all were marked.

Overall, the manuscript can be improved with consideration of the following comments.

Title & Abstract

  1. Page 1, Title: Suggest following for grammatical correctness “Feasibility of 3D-printed locking compression plates with Polyether ether ketone(PEEK) in tibial diaphyseal comminuted fractures”

Introduction

  1. Page 1, Line 10: Subject went from singular “PEEK” to a plural “they”. Suggest changing to singular “it”.
  2. Page 2, Line 3-4: Suggest adding the young’s modulus range of cortical bone for better clarity of PEEK comparisons
  3. Page 2, Line 13: Replace one of the uses of “traditional”
  4. Page 2, Line 33: Sentence is too wordy. Cut down or split into 2 sentences.
  5. Page 2, Line 50: Do not use “process” 3 times in one sentence so close to each other. An example of how this could be changed is “The benefit of the PEEK FDM compared to the SLS process is the highly stable manufacturing conditions and negligible shrinkage afterwards”

Methods

  1. Page 3, Line 13: More clarification needed on what “four symmetrically on each side” means
  2. Page 4, Line 4: Clarify what the melting point of the material is
  3. Page 4, Line 5: Clarify what the glass transition temperature of the material is
  4. Page 4, Line 22: “per-formed” should be performed
  5. Page 5, Figure 3, Line 2: Would be best to specify 3D-printed “PEEK LCP fixed tibia (right)”
  6. Need to divide the methods and results in the manuscript. All experimental and computational methods are detailed in the results section.
  7. Please provide experimental testing parameters including machine details, loading rates, fixation techniques, etc. Move testing procedure form Data & Results section to Methods section.

Results

  1. Page 5, Line 3: Should not have a comma before “three”
  2. Page 5, Line 13: Should not have a comma before “load-displacement”
  3. Page 6, Table 3: units for bending structural stiffness and bending strength should be (MN/mm2) and (N/mm) respectively
  4. Page 6, Line 8: wording should be changed so the sentences do not both end and immediately begin with “as shown in figure 5”
  5. Page 6, Line 8: Sentence starting with “As shown in figure 5” should not have comma following
  6. Page 6, Line 21: Very confusing wording of “...load of about 20% and a stiffness of about 45% compared to…”. Should be reworded as “...load of 20% of, and stiffness of 45% of that of the tibia structure…”. Also the word “about” should not be used as it takes makes the sentence sound less professional
  7. Page 7, Line 1: Specify whether the torque was applied at the top or the bottom of the fixed specimen
  8. Page 7, Line 3: “de-formation” should be deformation
  9. Page 7, Lines 2-4: Sentence should be reworded as “In the 3D-printed PEEK LCP no fracture occurred, only deformation, so torque was…”
  10. Page 7, Table 5: Units should be written as “N-m” to avoid confusion
  11. Page 8, Line 1-2: “by the” should be changed to “using an”
  12. Page 8, Line 14: “it is shown in Figure 7(a)” should be changed to “this setup is shown in Figure …” for grammatical clarity
  13. Page 8, Line 21: Should be changed to “... 0, 1 and 10 N-m increments” to be in numerical order
  14. Page 9, Line 4: missing a space in between number and unit (this mistake is repeated in following lines on this page) and Nm should be changed to N-m everywhere for clarity
  15. It is recommended that a table be included to directly compare experimental and FEA results.

Discussion/Limitations

  1. Page 9, Line 11: “extremely lower” should be changed to “significantly lower”
  2. Page 10, Line 9-12: Sentence is very confusing and should be reworded for clarity
  3. Page 10, Line 12: Remove “as a whole”
  4. Page 10, Line 20: Unclear what is meant by “... prepared through experiments under various conditions”. This should be reworded for clarity
  5. Page 10, Line 21: “through additional research” should be changed to “following additional research”
  6. How can this device be improved to potentially serve as a viable fixation device for these fractures?

Conclusion

  1. Page 10: Suggest not starting sentences with “However” and “Nevertheless”. They can be removed as they do not add anything to the sentences they are a part of.

Comments to Authors:

As is stated in the Discussion section, the main lamination of the study was the small sample size of only 2 test specimens. It would be best to bolster that with more analysis of the FEA data. There is no table present discussing the results of the FEA data, only the initial parameters.

This manuscript needs to be carefully edited for grammar. Many corrections can be found below, but not all were marked.

Overall, the manuscript can be improved with consideration of the following comments.

Title & Abstract

  1. Page 1, Title: Suggest following for grammatical correctness “Feasibility of 3D-printed locking compression plates with Polyether ether ketone(PEEK) in tibial diaphyseal comminuted fractures”

Introduction

  1. Page 1, Line 10: Subject went from singular “PEEK” to a plural “they”. Suggest changing to singular “it”.
  2. Page 2, Line 3-4: Suggest adding the young’s modulus range of cortical bone for better clarity of PEEK comparisons
  3. Page 2, Line 13: Replace one of the uses of “traditional”
  4. Page 2, Line 33: Sentence is too wordy. Cut down or split into 2 sentences.
  5. Page 2, Line 50: Do not use “process” 3 times in one sentence so close to each other. An example of how this could be changed is “The benefit of the PEEK FDM compared to the SLS process is the highly stable manufacturing conditions and negligible shrinkage afterwards”

Methods

  1. Page 3, Line 13: More clarification needed on what “four symmetrically on each side” means
  2. Page 4, Line 4: Clarify what the melting point of the material is
  3. Page 4, Line 5: Clarify what the glass transition temperature of the material is
  4. Page 4, Line 22: “per-formed” should be performed
  5. Page 5, Figure 3, Line 2: Would be best to specify 3D-printed “PEEK LCP fixed tibia (right)”
  6. Need to divide the methods and results in the manuscript. All experimental and computational methods are detailed in the results section.
  7. Please provide experimental testing parameters including machine details, loading rates, fixation techniques, etc. Move testing procedure form Data & Results section to Methods section.

Results

  1. Page 5, Line 3: Should not have a comma before “three”
  2. Page 5, Line 13: Should not have a comma before “load-displacement”
  3. Page 6, Table 3: units for bending structural stiffness and bending strength should be (MN/mm2) and (N/mm) respectively
  4. Page 6, Line 8: wording should be changed so the sentences do not both end and immediately begin with “as shown in figure 5”
  5. Page 6, Line 8: Sentence starting with “As shown in figure 5” should not have comma following
  6. Page 6, Line 21: Very confusing wording of “...load of about 20% and a stiffness of about 45% compared to…”. Should be reworded as “...load of 20% of, and stiffness of 45% of that of the tibia structure…”. Also the word “about” should not be used as it takes makes the sentence sound less professional
  7. Page 7, Line 1: Specify whether the torque was applied at the top or the bottom of the fixed specimen
  8. Page 7, Line 3: “de-formation” should be deformation
  9. Page 7, Lines 2-4: Sentence should be reworded as “In the 3D-printed PEEK LCP no fracture occurred, only deformation, so torque was…”
  10. Page 7, Table 5: Units should be written as “N-m” to avoid confusion
  11. Page 8, Line 1-2: “by the” should be changed to “using an”
  12. Page 8, Line 14: “it is shown in Figure 7(a)” should be changed to “this setup is shown in Figure …” for grammatical clarity
  13. Page 8, Line 21: Should be changed to “... 0, 1 and 10 N-m increments” to be in numerical order
  14. Page 9, Line 4: missing a space in between number and unit (this mistake is repeated in following lines on this page) and Nm should be changed to N-m everywhere for clarity
  15. It is recommended that a table be included to directly compare experimental and FEA results.

Discussion/Limitations

  1. Page 9, Line 11: “extremely lower” should be changed to “significantly lower”
  2. Page 10, Line 9-12: Sentence is very confusing and should be reworded for clarity
  3. Page 10, Line 12: Remove “as a whole”
  4. Page 10, Line 20: Unclear what is meant by “... prepared through experiments under various conditions”. This should be reworded for clarity
  5. Page 10, Line 21: “through additional research” should be changed to “following additional research”
  6. How can this device be improved to potentially serve as a viable fixation device for these fractures?

Conclusion

  1. Page 10: Suggest not starting sentences with “However” and “Nevertheless”. They can be removed as they do not add anything to the sentences they are a part of.

Author Response

Reviewer 3

As is stated in the Discussion section, the main lamination of the study was the small sample size of only 2 test specimens. It would be best to bolster that with more analysis of the FEA data. There is no table present discussing the results of the FEA data, only the initial parameters.

This manuscript needs to be carefully edited for grammar. Many corrections can be found below, but not all were marked.

Overall, the manuscript can be improved with consideration of the following comments.

Title & Abstract

Page 1, Title: Suggest following for grammatical correctness “Feasibility of 3D-printed locking compression plates with Polyether ether ketone(PEEK) in tibial diaphyseal comminuted fractures”

  • Thanks to reviewer’s comment. We modified the title of the paper according to the reviewer’s suggestion.

Introduction

Page 1, Line 10: Subject went from singular “PEEK” to a plural “they”. Suggest changing to singular “it”.

  • Thanks to reviewer’s comment. We modified it according to the reviewer’s suggestion.

Page 2, Line 3-4: Suggest adding the young’s modulus range of cortical bone for better clarity of PEEK comparisons

  • Thanks to reviewer’s comment. We added the range of Young’s modulus of cortical bone into the texts.

Page 2, Line 13: Replace one of the uses of “traditional”

  • Thanks to reviewer’s comment. We corrected it.

Page 2, Line 33: Sentence is too wordy. Cut down or split into 2 sentences.

  • Thanks to reviewer’s comment. We corrected 3 sentences according to reviewer’s comment.

Page 2, Line 50: Do not use “process” 3 times in one sentence so close to each other. An example of how this could be changed is “The benefit of the PEEK FDM compared to the SLS process is the highly stable manufacturing conditions and negligible shrinkage afterwards”

  • Thanks to reviewer’s comment. We corrected the sentence according to reviewer’s comment.

Methods

Page 3, Line 13: More clarification needed on what “four symmetrically on each side” means

  • Thanks to reviewer’s comment. We modified it just as below.

The eight holes were symmetrically placed in groups of four around the center of the LCP.

Page 4, Line 4: Clarify what the melting point of the material is

  • Thanks to reviewer’s comment. We specified the name of material and its melting point.

Page 4, Line 5: Clarify what the glass transition temperature of the material is

  • Thanks to reviewer’s comment. We specified the name of material and its glass transition temperature.

Page 4, Line 22: “per-formed” should be performed

  • Thanks to reviewer’s comment. We corrected it.

Page 5, Figure 3, Line 2: Would be best to specify 3D-printed “PEEK LCP fixed tibia (right)”

  • Thanks to reviewer’s comment. We corrected it.

Need to divide the methods and results in the manuscript. All experimental and computational methods are detailed in the results section.

Please provide experimental testing parameters including machine details, loading rates, fixation techniques, etc. Move testing procedure form Data & Results section to Methods section.

  • Thanks to reviewer's comment. As with the reviewer's comment, we confirmed that there was a section describing the methods for the test in "Data & Results". Therefore, “2.3. Methods for mechanical tests” was added and the test methods described in “Data & Results” were moved to the section.

Results

Page 5, Line 3: Should not have a comma before “three”

  • Thanks to reviewer’s comment. We re-write the sentence to avoid using comma before “three”.

Page 5, Line 13: Should not have a comma before “load-displacement”

  • Thanks to reviewer’s comment. We re-write the sentence to avoid using comma before “load-displacement”.

Page 6, Table 3: units for bending structural stiffness and bending strength should be (MN/mm2) and (N/mm) respectively

  • Thanks to point out our mistakes. We corrected them.

Page 6, Line 8: wording should be changed so the sentences do not both end and immediately begin with “as shown in figure 5”, Page 6, Line 8: Sentence starting with “As shown in figure 5” should not have comma following

  • Thanks to reviewer’s comment. We re-write the sentences.

Page 6, Line 21: Very confusing wording of “...load of about 20% and a stiffness of about 45% compared to…”. Should be reworded as “...load of 20% of, and stiffness of 45% of that of the tibia structure…”. Also the word “about” should not be used as it takes makes the sentence sound less professional

  • Thanks to reviewer’s suggestion. We corrected it according to reviewer’s suggestion.

Page 7, Line 1: Specify whether the torque was applied at the top or the bottom of the fixed specimen

  • Thanks to reviewer’s comment. In the section “Methods for Mechanical Test”, we corrected the sentence describing condition for torsional test to involve information about it.

Page 7, Line 3: “de-formation” should be deformation

  • Thanks to reviewer’s comment. We corrected it.

Page 7, Lines 2-4: Sentence should be reworded as “In the 3D-printed PEEK LCP no fracture occurred, only deformation, so torque was…”

  • Thanks to reviewer’s comment. We corrected the sentence according to reviewer’s comment.

Page 7, Table 5: Units should be written as “N-m” to avoid confusion

  • Thanks to reviewer’s comment. We corrected it.

Page 8, Line 1-2: “by the” should be changed to “using an”

  • Thanks to reviewer’s comment. We corrected it.

Page 8, Line 14: “it is shown in Figure 7(a)” should be changed to “this setup is shown in Figure …” for grammatical clarity

  • Thanks to reviewer’s comment. We corrected it according to reviewer’s comment.

Page 8, Line 21: Should be changed to “... 0, 1 and 10 N-m increments” to be in numerical order

  • Thanks to reviewer’s comment. We corrected it according to reviewer’s comment.

Page 9, Line 4: missing a space in between number and unit (this mistake is repeated in following lines on this page) and Nm should be changed to N-m everywhere for clarity

  • Thanks to reviewer’s comment. We inserted a space in between number and unit, and all typo on N-m were corrected.

It is recommended that a table be included to directly compare experimental and FEA results.

  • Thanks to reviewer’s comment. According to reviewer’s suggestion, we inserted a table that directly compare simulated results with experimental results.

Discussion/Limitations

Page 9, Line 11: “extremely lower” should be changed to “significantly lower”

  • Thanks to reviewer’s comment. We corrected it according to reviewer’s comment.

Page 10, Line 9-12: Sentence is very confusing and should be reworded for clarity

  • Thanks to reviewer’s comment. We divided the sentence into two sentences to clarify meaning.

Page 10, Line 12: Remove “as a whole”

  • Thanks to reviewer’s comment. We removed it.

Page 10, Line 20: Unclear what is meant by “... prepared through experiments under various conditions”. This should be reworded for clarity. Page 10, Line 21: “through additional research” should be changed to “following additional research”

  • Thanks to reviewer’s comment. The contents of the paragraph involve commented sentences are judged to be unnecessary to understand the result of our study. Therefore, that paragraph has been deleted. In conclusion, we concluded that PEEK LCP fabricated through FDM 3D printing is not suitable for the tibial non-osteoporotic diaphyseal comminuted fracture condition. And, we also suggested that PEEK LCP fabricated through FDM 3D printing can be applicable under conditions such as non-weight-bearing region (clavicle, humerus, forearm bone and so on) fracture where small load is applied.

How can this device be improved to potentially serve as a viable fixation device for these fractures?

  • In conclusion, we concluded that PEEK LCP fabricated through FDM 3D printing is not suitable for the tibial non-osteoporotic diaphyseal comminuted fracture condition. And, we also suggested in “Conclusion” that PEEK LCP fabricated through FDM 3D printing can be applicable under conditions such as non-weight-bearing regions fracture where small load is applied.

Conclusion

Page 10: Suggest not starting sentences with “However” and “Nevertheless”. They can be removed as they do not add anything to the sentences they are a part of.

  • Thanks to reviewer’s comment. The contents of the paragraph involve commented sentences are judged to provoke confusing. So, we deleted the sentence.

Round 2

Reviewer 2 Report

After carefully check the entire manuscript, its only 1 issue is not solved.

  1. This claim “Thus, while 3D-printed PEEK LCP is not suitable for tibial non-osteoporotic diaphyseal comminuted fracture conditions, it may be used for fracture conditions in non-weight-bearing regions.” If this study is not suitable, then, it is valid or it is have novelty for the present study?

Please revised this part or paraphare it so that the negative information related to the not suitable can be erased. After this issue fixed, I think the paper can be accepted.

Author Response

Reviewer 2

  1. This claim “Thus, while 3D-printed PEEK LCP is not suitable for tibial non-osteoporotic diaphyseal comminuted fracture conditions, it may be used for fracture conditions in non-weight-bearing regions.” If this study is not suitable, then, it is valid or it is have novelty for the present study?
  • Thank you for the reviewer's suggestions. We agree that including a sentence in the abstract that undermines the originality of the study could potentially divert the interest of researchers who are interested in the content of this study. Therefore, I have rewritten the content in the abstract to state that the 3D printed PEEK LCP produced through this study can be applied to fracture fixation surgeries in non-weight bearing regions.
  • Furthermore, this study has the following originality and strengths

One of our study’s strengths is that we made the PEEK LCP using 3D-print and compared the commercial LCP plate and 3D-printed PEEK LCP under various conditions in the specific tibial non-osteoporotic diaphyseal fracture. Many studies have been reported on surgical implants made of 3D printed PEEK. However, there was no study comparing commercial plate and PEEK plate under specific condition such as tibial non-osteoporotic fracture.

Based on our research findings, the significant strength of our study is the prospect that the 3D-printed PEEK LCP might potentially replace the commercial LCP plate currently used in clinical practice for fracture conditions in non-weight-bearing regions such as upper extremities. We clearly stated about this results in the “Conclusion”.

Furthermore, because of less previous studies, the combination of a biomechanical study using saw bone and finite element analysis in our research is regarded valuable.

Reviewer 3 Report

Comments to Authors:

            The flow of the paper is much better following the improvements. Structure and syntax make a lot more sense and the results reflect the experiment much better.

Introduction

  1. Page 1, Line 11: Change to “Further, because it is a bio-inactive material, it doesn’t cause…”

Author Response

(The authors gave the same response as above.)
